# Prospective study of factors associated with asthma attack recurrence (ATTACK) in children from three Ecuadorian cities during COVID-19: a study protocol

Diana Morillo,[1] Santiago Mena-Bucheli,[1] Angélica Ochoa,[2] Martha E Chico,[1] Claudia Rodas,[3] Augusto Maldonado,[4,5] Karen Arteaga,[6] Jessica Alchundia,[7] Karla Solorzano,[7] Alejandro Rodriguez,[1] Camila Figueiredo,[8] Cristina Ardura-Garcia,[9] Max Bachmann  ,[10] Michael Richard Perkin  ,[11] Irina Chis Ster  ,[12,12] Alvaro Cruz,[13] Natalia Cristina Romero  ,[14,15] Philip Cooper  [1,12]

NCR and PC contributed equally.

For numbered affiliations see end of article.

**Correspondence to**
Professor Philip Cooper;
pcooper@sgul.ac.uk

## ABSTRACT

**Introduction** Asthma is a growing health problem in children in marginalised urban settings in low-income and middle-income countries. Asthma attacks are an important cause of emergency care attendance and long-term morbidity. We designed a prospective study, the Asthma Attacks study, to identify factors associated with recurrence of asthma attacks (or exacerbations) among children and adolescents attending emergency care in three Ecuadorian cities.

**Methods and analysis** Prospective cohort study designed to identify risk factors associated with recurrence of asthma attacks in 450 children and adolescents aged 5–17 years attending emergency care in public hospitals in three Ecuadorian cities (Quito, Cuenca and Portoviejo). The primary outcome will be rate of asthma attack recurrence during up to 12 months of follow-up. Data are being collected at baseline and during follow-up by questionnaire: sociodemographic data, asthma history and management (baseline only); recurrence of asthma symptoms and attacks (monthly); economic costs of asthma to family; Asthma Control Test; Pediatric Asthma Quality of life Questionnaire; and Newcastle Asthma Knowledge Questionnaire (baseline only). In addition, the following are being measured at baseline and during follow-up: lung function and reversibility by spirometry before and after salbutamol; fractional exhaled nitric oxide (FeNO); and presence of IgG antibodies to SARS-CoV-2 in blood. Recruitment started in 2019 but because of severe disruption to emergency services caused by the COVID-19 pandemic, eligibility criteria were modified to include asthmatic children with uncontrolled symptoms and registered with collaborating hospitals. Data will be analysed using logistic regression and survival analyses.

**Ethics and dissemination** Ethical approval was obtained from the Hospital General Docente de Calderon (CEISH-HGDC 2019-001) and Ecuadorian Ministry of Public Health (MSP-CGDES-2021-0041-O N° 096-2021). The study results will be disseminated through presentations at conferences and to key stakeholder groups including policy-makers, postgraduate theses, peer-review publications and a study website. Participants gave informed consent to participate in the study before taking part.

## Strengths and limitations of this study

⇒ Asthma attacks are a growing cause of morbidity in children and burden on health systems in low-income and middle-income countries, and there is limited information on factors associated with asthma attacks in poor urban settings in Latin America.

⇒ This protocol describes a study designed to investigate risk factors associated with recurrence of asthma attacks in children and adolescents living in marginalised settings in three Ecuadorian cities.

⇒ This study is recruiting children and adolescents with uncontrolled asthma symptoms and asthma attacks and following them prospectively for 6–12 months to identify factors associated with recurrence of attacks in the context of the COVID-19 pandemic.

⇒ The COVID-19 pandemic has had a major impact on health-seeking behaviours, while mitigation strategies to control transmission of the novel coronavirus, SARS-CoV-2, have resulted in marked declines in the circulation of respiratory viruses that are considered to underlie a high proportion of asthma attacks. Such factors may interfere with recruitment and follow-up of study participants.

## INTRODUCTION

Asthma is the most common chronic disease of childhood and is estimated to affect more than 350 million people worldwide.[1]

Although initially described as a disease of high-income countries (HICs) such as the UK, international comparisons using standardised methodologies have shown a similar or greater prevalence among children in urban centres in Latin America as observed in HICs.[2] There is evidence that the prevalence of asthma has increased in low-income and middle-income countries (LMICs) over recent decades.[2]

Asthma attacks or exacerbations are frequent in children and are a common cause of hospitalisation,[3] most frequently associated with respiratory viral infections,[4] but also with exposures to allergens[5] and air pollution[6] among individuals with an underlying genetic susceptibility.[7] Asthma is an underdiagnosed disease because of lack of specialised personnel and resources to evaluate lung function. Further, limited access to specialised care and treatment,[8] accompanied by poor treatment adherence[9 10] are important contributing factors to inadequate disease control and attacks, especially in underprivileged populations. In such settings, asthma patients with poor control of their daily symptoms often rely on emergency room (ER) care for management of their symptoms,[11] resulting in high economic costs to health systems and patients' families.[8 12]

To understand better factors contributing to the risk of asthma attacks, it is important to identify those factors contributing to suboptimal management, which can be improved through better healthcare.[13] Several studies have investigated potential predictors of recurrence of asthma attacks among children attending emergency services. Factors identified as important included a history of previous emergency attendances for acute asthma attacks, younger age,[13–15] ethnicity of African descent[16 17] and low socioeconomic status.[13 18] These studies were conducted almost exclusively in North America and frequently included adult Hispanic populations living there. There are relatively few such studies done in LMICs.[13 16 18]

The Asthma Attacks Study (ATTACK) aims to identify factors related to recurrence of asthma attacks in Ecuadorian children attending emergency care in three Ecuadorian cities. Recruitment into the study started in March 2019, but was interrupted by the COVID-19 pandemic that had a major impact on use of and access to emergency care.[19] In consequence, the original protocol was modified to allow continued recruitment and measure the impact of the COVID-19 pandemic and exposures to SARS-CoV-2 on asthma attacks in children and adolescents.

### What is known about factors related to asthma attacks and impact of COVID-19 pandemic?

Severe asthma in children is associated with loss of lung function,[20] as well as with high costs of medical and family care.[2] In Latin America, asthma has been described as a major public health problem because of a high prevalence and significant associated morbidity and mortality.[21] Asthma control among children in Latin America has

been reported to be among the worst worldwide.[22–25] The Latin America Asthma Insights and Management Study reported adequate asthma control in less than 20% of asthmatic children in Argentina, Brazil, Mexico, Venezuela and Puerto Rico.[22] A case–control study of asthmatic children with an acute attack attending an ER in coastal Ecuador showed that, over the previous 12 months, only 20% of children had visited a doctor in the past year for a routine visit, 86% had attended an ER at least once prior and none were receiving inhaled corticosteroids (ICS) despite over half having suffered four or more acute exacerbations.[26] A previous prospective study of children with asthma attacks in coastal Ecuador showed a recurrence rate of 46% within 6 months.[13]

Asthma attacks represent an acute or subacute increase in respiratory symptoms[25] and severe attacks—defined as requiring ER or hospital care and treatment with systemic corticosteroids for at least 3 days—are associated with loss of lung function[20] and high economic costs to the patient's family and healthcare system,[7] as well as absenteeism in work and school.

Asthma attacks may be prevented either by avoiding previously identified triggers or by appropriate preventive treatment. Patients using ICS were three times more likely to have well-controlled asthma,[22] a reduction in symptoms independent of frequency, and better lung function.[1] In Brazil, preventive strategies have been successfully implemented through asthma control programmes such as Pro-AR (Programa de Controle da Asma e da Rinite Alérgica na Bahia) in Salvador de Bahía to reduce future risk in asthmatic patients from less privileged backgrounds, through the provision of ICS.[27] There has been a shift to using ICS more widely for control of asthma, for example, through the provision of combination inhalers including both ICS and a beta$_2$ agonist for moderate-to-severe asthma.[28] More recent evidence indicates that the use of ICS may be beneficial even for asthmatic children with mild disease that accounts for 30%–40% of severe attacks.[29] Treatment recommendations for children and adolescents with mild disease are now shifting to use of ICS (plus long-acting beta-agonists in combination inhalers) as required.[30 31] Despite such initiatives, access to ICS in many LMIC settings is limited within healthcare systems because of cost and a failure to recognise the public health impact of childhood asthma. In such settings, children with asthma symptoms frequently over-rely on ER care for the acute control of their symptoms in the absence of adequate provision of specialised care for their long-term management.[26 32]

Soon after the emergence of the novel coronavirus, SARS-CoV-2, in humans and its spread into a global pandemic, the question was raised as to whether this novel viral infection might alter the risk of asthma symptoms because of deficient antiviral immune responses and the tendency to increase exacerbations during infections with other respiratory viruses.[33] To date, evidence from studies of asthmatic children and adults do not indicate an increased risk.[34 35] Characteristics of asthmatic airways

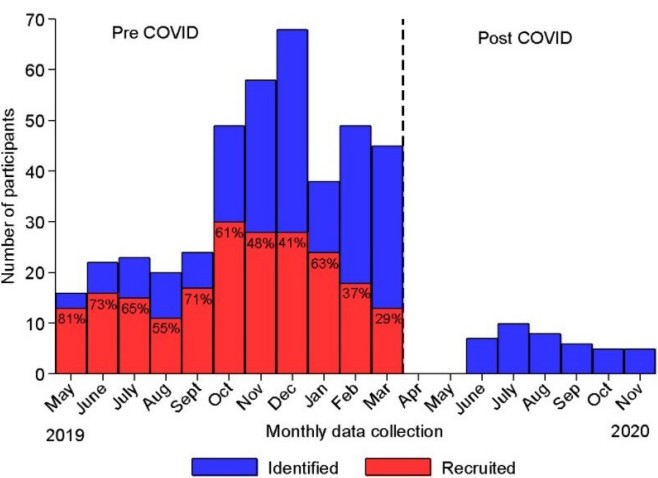

**Figure 1** Impact of the COVID-19 pandemic in the three study centres in Quito, Portoviejo and Cuenca, Ecuador, on recruitment rates into the study and number of potentially eligible subjects identified in collaborating emergency rooms. Red bars show numbers recruited as a proportion (%) of those eligible (blue bars) prior to a legally enforced national lockdown on 16 March 2020.

among those with T2-linked phenotypes may reduce the expression of the cellular receptor for SARS-CoV-2, ACE2, in the airways.[36] It has been suggested that the use of ICS among asthmatics may reduce risk of severe COVID-19 by suppressing inflammation and enhancing antiviral defences,[37] although the evidence remains unclear.[38] There are limited data from paediatric populations.[38] The COVID-19 pandemic has resulted in marked reductions in hospital visits including to ERs (figure 1), reduced attack rates and better lung function among children with asthma,[19 39] which are likely consequences of mitigation measures to reduce transmission of SARS-CoV-2 (such as lockdowns, face masks and social distancing) that have resulted also in the reduced transmission of other respiratory viruses.[4 6 10 39] Treatment adherence may have improved also because of better parental supervision of asthma medications.[40] There are limited data on the impact of exposure to SARS-CoV-2 on asthma attacks and uncontrolled symptoms and the impact of the collateral effects of the COVID-19 pandemic including reduced access to health services and medications and effects of COVID-19 mitigation efforts in resource-poor LMIC settings.

## METHODS AND ANALYSIS
### The Asthma Attack Study
A prospective multicentre study is being done in children aged 5–17 years in low-income settings in three Ecuadorian cities (Quito, Cuenca and Portoviejo) to study factors associated with acute asthma attacks. Because of the impact of the COVID-19 pandemic on risk of asthma attacks and healthcare access, the study was divided into two phases of recruitment and follow-up: (1) phase I (or pre-COVID-19 from May 2019 to March 2020) evaluated

risk factors associated with recurrence of acute asthma attacks among children and adolescents attending ERs of public hospitals in low-income settings in the three cities and (2) phase II (or peri-COVID-19 from March 2021 to June 2022) is evaluating risk factors associated with attacks among children and adolescents with asthma and current symptoms who have previously attended specialist care at public hospitals in Cuenca and Portoviejo. Phase II is also evaluating the effects of exposures to SARS-CoV-2 on asthma attacks and symptoms. In addition, both study phases are evaluating the impact of asthma symptoms and adequate control of asthma on the quality of life and economic costs for the family of children and adolescents with asthma. The objectives of phases I and II are detailed in table 1. Qualitative studies of health worker's perspectives on asthma care coordination between primary and specialised healthcare are being done also as described elsewhere.[41]

### Study design and setting
This is a multicentre prospective study conducted in three Ecuadorian cities (Quito, Cuenca and Portoviejo). Quito, the capital, and Cuenca, the third largest city, are located in the Andean highlands (altitude>2500 m) where the annual average temperature is ~16°C, while Portoviejo is in the coastal region (at sea level) with an annual average temperature of ~25°C.[42] According to the 2010 census, the populations of the three cities are: Quito>2 million, Cuenca 0.5 million and Portoviejo 0.28 million inhabitants. Ecuador is an upper-middle-income country with a per capita income of US$6080 in 2019. The health system in Ecuador includes public and private provision. Public institutions offer universal health coverage that is stratified into different levels of care from primary to tertiary services.[43] Social security institutions offer health services to affiliated salaried workers and their families. The private sector includes for-profit entities (hospitals, clinics, dispensaries, doctor's offices, pharmacies and prepaid health insurance companies), which are generally located in the main cities.[44]

### Study centers
The study was initially based in tertiary care hospitals in each of the three cities (Hospital Docente General de Calderon, Quito; Hospital Vicente Corral Moscoso, Cuenca; and Hospital Verdi Cevallos Balda, Portoviejo).

### Recruitment: phase I (pre-COVID-19; May 2019 to March 2020)
Children and adolescents aged 5–17 years were recruited either while attending ERs with an asthma attack or through daily registries and contact telephone numbers of patients attending the ERs with an asthma attack. From May 2019, eligible patients (table 2) were provided with a study information sheet on discharge and their parents were contacted to arrange an interview at a hospital consulting room 10–14 days later when they were recruited into the study after providing informed written parental consent and minor assent (8 years and older).

**Table 1** Study objectives for phases I and II, how each objective will be achieved, and statistical analysis for each objective

| Phase | Objectives | How objective will be achieved | Statistical outcome | Statistical methods | Statistical inference |
|---|---|---|---|---|---|
| 1 | **Primary** | | | | |
| | Risk factors associated with asthma attacks recurrence requiring an emergency visit during follow-up | Monthly follow-up and recording of events | Binary: yes/no indicating at least one individual recurrence | Logistic regression | OR and 95% CIs; predictive model for recurrence, ROC and AUC |
| | **Secondary** | | | | |
| | 1. Risk factors associated with time to first asthma attack recurrence | Monthly follow-up and recording of events | Binary: yes/no indicating the first recurrence and/or time to the first event | Survival analyses modelling time to first event | HR and 95% CIs; survival models—proportional hazard type (semiparametric Cox or parametric Weibull) or accelerated time failure depending on data |
| | 2. Risk factors associated with monthly asthma attack recurrence | Monthly follow-up and recording of events | Longitudinal binary outcomes indicating events and times | Longitudinal binary outcome | OR; HR and 95% CIs; longitudinal binary models (GEE) Multiple events per individual - frailty survival analysis |
| | 3. Evaluate impact of asthma recurrence and control on quality of life and economic costs for patient's families | Monthly follow-up and recording of events and questionnaire on asthma control (0, 6 and 12 months) and quality of life and economic costs at 6 and 12 months | Longitudinal continuous outcomes indicating scores for quality of life and economic costs | Longitudinal continuous data outcomes and time-varying covariates | Longitudinal continuous data analysis with time varying binary covariates indicating presence of recurrent events |
| 2 | **Primary** | | | | |
| | Risk factors associated with asthma attacks recurrence requiring an emergency visit during follow-up | Monthly follow-up and recording of events | Binary: yes/no indicating at least one individual recurrence during follow-up | Logistic regression | OR and 95% CIs; predictive model, ROC, AUC |
| | **Secondary** | | | | |
| | 1. Risk factors associated with time to first asthma attack recurrence | Monthly follow-up and recording of events | Binary: yes/no indicating the first recurrence and/or time to first event | Survival type analyses modelling time to first event | HR and 95% CIs; survival models—proportional hazard type (semiparametric Cox or parametric Weibull) or accelerated time failure depending on data |
| | 2. Risk factors associated with monthly asthma attack recurrence | Monthly follow-up and recording of events | Longitudinal binary outcomes indicating events and times | Longitudinal binary outcome | OR; HR and 95% CIs; longitudinal binary models (GEE estimation) Multiple events per individual—frailty survival analysis |
| | 3. Evaluate impact of asthma recurrence and control on quality of life and economic costs for patient's families | Monthly follow-up and recording of events and questionnaire on asthma control (0, 6 and 12 months) and quality of life and economic costs at 6 and 12 months | Longitudinal continuous outcomes indicating scores for quality of life and economic costs | Longitudinal continuous outcomes and time-varying covariates | OR and 95% CIs; longitudinal continuous data analysis with time varying binary covariates indicating presence of recurrent events |
| | 4. Effects of seropositivity to SARS-CoV-2 on risk of any asthma attack recurrence and number of events | Serology for SARS-CoV-2 at 0 and 6 months | Longitudinal binary outcomes indicating events and times | Longitudinal binary outcome and time-varying covariates | Longitudinal binary analysis with time varying binary indicating SARS-CoV-2 serology. Both population averages and subject-specific models to be considered given time-varying nature of the SARS-CoV-2 variable |

AUC, area under the curve; GEE, generalised estimating equations; ROC, receiving operator characteristic.

## Recruitment: phase II (peri-COVID-19; March 2021 to March 2022)

The COVID-19 pandemic had a dramatic impact on our ability to recruit children with asthma attacks in the three study hospitals that resulted in the suspension of recruitment in March 2020. This was owing to the reassignment of all three hospitals to COVID-19-only activities for prolonged periods and the imposition of a

**Table 2** Eligibility criteria for entry into phase I and phase II of study

| Characteristic | Phase I | Phase II |
|---|---|---|
| Setting/study population | Public hospital ERs in Quito, Cuenca and Portoviejo | Public hospital registry of patients with asthma in Cuenca and Portoviejo |
| Inclusion criteria | 1. Aged 5–17 years<br>2. Acute asthma attack attending ERs at public hospitals<br>3. Living within 12 km of public hospital<br>4. Informed written consent from parents<br>5. Minor assent from children≥8 years | 1. Aged 5–17 years<br>2. Wheeze within the last 6 months<br>3. Living within 12 km of public hospital<br>4. Informed written consent from parents<br>5. Minor assent from children≥8 years |
| Exclusion criteria | 1. Other chronic disease<br>2. Living>12 km from public hospital | 1. Other chronic disease<br>2. Living>12 km from public hospital |

ERs, emergency rooms.

legally enforced lockdown nationally. Figure 2 shows the number of potentially eligible children identified in the three study hospitals by month in the period prior to COVID-19 and during the pandemic. There was a marked reduction in eligible children attending the ERs immediately following the imposition of lockdown on 16 March 2020 and during the following 8 months. In the period following the initial lockdown and other mitigation strategies to reduce transmission of SARS-CoV-2, within the sample of asthmatic children recruited into phase I of the study, there was evidence of dramatic reduction in health service attendance rather than changes in the incidence of attacks.[19] To allow continued recruitment, a second phase (phase II) of the study was started with eligibility criteria modified to include children and adolescents aged 5–17 years registered as having asthma in patient registries at the study hospitals in Cuenca and Portoviejo (Quito was not included in phase II because of logistic issues in that specific setting). The child's parent was contacted by telephone and asked about the child's current asthma

symptoms. Children with current symptoms were considered eligible. Eligibility criteria for phases I and II are shown in table 2.

## Data collection
A study manager supervised day-to-day activities of dedicated teams (physician and nurse) of trained study personnel based in each of the public hospitals. Each participant was followed up for 12 months in phase I and at least 6 months in phase II. Standardised procedures were used throughout (figure 2). Subjects were evaluated at monthly intervals using face-to-face and telemonitoring for follow-up. Both phases include face-to-face evaluations (evaluations at baseline, 2, 4, 6, 9 and 12 months in phase I; baseline, 6 and 12 months in phase II). Between March 2020 and March 2021, all face-to-face evaluations in phase I were changed to telemonitoring apart from the final evaluation at 12 months, which remained face-to-face, where possible. Final face-to-face evaluations were done in March–April 2021 on participants who had completed 12 months of follow-up between June 2019 and February 2020. The site of face-to-face evaluations was changed to either a clinic or home visit depending on the preference of the child's parents. Study procedures were the same in both phases of the study (figure 2) with the following exceptions: (1) procedures requiring face-to-face evaluations at 6 and 12 months in phase I (Childhood Asthma Control Test (C-ACT) and Pediatric Asthma Quality of Life Questionnaire (PAQLQ)) were dropped in phase II and the remaining instruments were administered by telemonitoring; (2) blood samples (baseline, 6 and 12 months) are being collected from those recruited into phase II of the study for measurement of anti-SARS-CoV-2 IgG antibodies; and (3) measurement of lung function and fractional exhaled nitric oxide (FeNO) was suspended from March 2020 because of risks from potentially hazardous infectious aerosols until a time when this procedure can be done safely. Medications (inhaled salbutamol and fluticasone) were provided free of charge to participants when prescribed by a hospital physician and there was no medication in the hospital pharmacy. The study team ensured adequate inhaler technique during

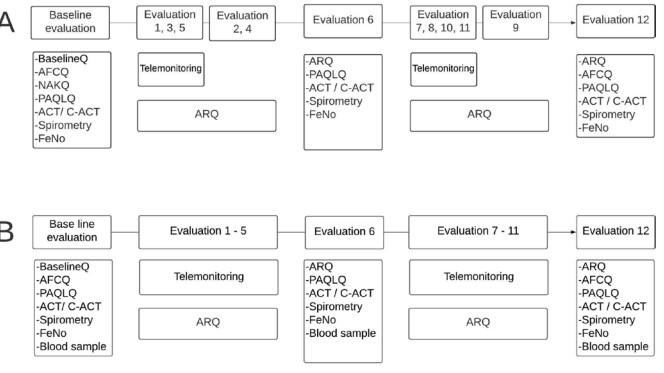

**Figure 2** Study procedures during baseline evaluation and follow-up in phases I (A) and II (B) of study. Follow-up in phase I was for 12 months and in phase II for a minimum of 6 months. ACT, Asthma Control Questionnaire (>11 years); AFCQ, Asthma Family Costs Questionnaire; ARQ, Asthma Recurrence Questionnaire; BaselineQ, general questionnaire based on phase II of the International Study of Asthma and Allergies in Childhood; C-ACT, Child Asthma Control Questionnaire (≥12 years); FeNO, fractional exhaled nitric oxide; NAKQ, Newcastle Asthma Knowledge Questionnaire.

**Table 3** Data collection at baseline and during follow-up in phases I and II of study

| Data collected | Baseline | Month 6 | Month 12 | Monthly |
|---|---|---|---|---|
| Asthma diagnosis | I/II | | | |
| History of asthma symptoms | I/II | | | |
| Current asthma symptoms | I/II | I/II | I/II | I/II |
| ER visits/hospitalisations | I/II | I/II | I/II | I/II |
| Asthma medications | I/II | I/II | I/II | I/II |
| Asthma control (C-ACT/ACT) | I/II | (I)/II | (I)/II | |
| Asthma quality of life (PAQLQ) | I/II | (I)/II | (I)/II | |
| Asthma knowledge (NAKQ) | I | | | |
| Lung function | I/(II) | (I/II) | (I) | |
| Reversibility | I/(II) | (I/II) | (I) | |
| FeNO | I/II | (I)/II | (I)/II | |
| Anti-SARS-CoV-2 IgG antibodies | II | II | II | |
| COVID-19 symptoms or diagnosis | II | II | II | II |

() represents data collection where possible.

(), Data collection where possible; C-ACT, Childhood Asthma Control Test; ER, emergency room; FeNO, fractional exhaled nitric oxide; NAKQ, Newcastle Asthma Knowledge Questionnaire; PAQLQ, Pediatric Asthma Quality of Life Questionnaire.

face-to-face clinic or home visits. Study personnel did not interfere with treatment indications of hospital staff.

### Questionnaires at baseline and follow-up

The following questionnaires were used as indicated in figure 2; (1) asthma questionnaire to collect information on history of asthma symptoms, treatment and management, and potential risk factors for asthma attacks. This questionnaire has been previously adapted from phase II of the International Study of Asthma and Allergies in Children, and extensively field tested for children and adolescents with asthma[13 26]; (2) Asthma Recurrence Questionnaire—questionnaire to monitor recurrence of asthma symptoms and monitor treatment during follow-up,[13] including symptoms and diagnosis of COVID-19 in phase II; (3) C-ACT,[45] completed by the child and parent for children aged<12 years or the ACT[46] for those aged≥12 years; (4) PAQLQ[47]; (5) Newcastle Asthma Knowledge Questionnaire[48 49] validated in a Spanish-speaking population; and (4) Asthma Family Cost Questionnaire.[50] Data collection is summarised in table 3.

### Lung function test and FeNO

Lung function testing is being done according to American Thoracic Society criteria using a portable ultrasonic spirometer (EasyOne device, ndd Medical Technologies). Global Lung Function Initiative 2012 reference values will be used to calculate z scores depending on age, sex, height and ethnicity.[51] Lung function is being measured before and after 200 µg salbutamol administered through a spacer with a positive test for airways reactivity defined as an increase in forced expiratory volume in one second of ≥12%. FeNO is being measured in parts per billion using NObreath (Bedfont Scientific, UK). As a consequence of lockdown and mitigation strategies to prevent transmission of SARS-CoV-2, measurement of lung function and FeNO was suspended in March 2020. FeNO measurement, a low effort procedure that carries a minimal risk of generating aerosols,[52 53] was resumed in March 2021 for 12-month evaluations in phase I and baseline evaluations in phase II. FeNO is being done following the recommendations of the manufacturer to minimise potential risks.[54] Spirometry, which often induces cough, will be resumed, only if the procedure can be done without risk to staff, patients and their families.

### Blood samples

Blood sample are being taken at baseline, 6 and 12 months in phase II. Blood is drawn from an antecubital vein into a 4 mL plastic tube containing EDTA (ethylenediaminetetraacetic acid) as anticoagulant (Vacutainer, BD). After centrifugation, plasma is stored at −20°C until analysis. Levels of IgG specific for SARS-CoV-2 are being measured using a validated enzyme linked immunosorbent assay.[55]

### Biosafety precautions relating to the COVID-19 pandemic

Personnel gathering data during face-to-face visits have been trained to use appropriate personal protective equipment according to international guidelines.[56–58] Immediately prior to a home or clinic visit, the child's carer is telephoned about potential symptoms or a COVID-19 diagnosis among family members or close contacts—if any symptoms are reported the visit is rescheduled after 2 weeks.

### Sample size considerations

A sample size of 450 patients was proposed for phase I, based on a binary outcome variable of any recurrence

of an asthma attack within 12 months or so. We used an estimated recurrence risk of about 50% during 6-month follow-up from a similar population in Ecuador.[13] The study should collect between 10 and 16 outcome events per each potential explanatory variable for robust results (ie, for at least 80% power, equivalent to a type 2 error of less than 20% as proven by simulation studies). Assuming 10 explanatory variables are considered, in the worst-case scenario, 10×16=160 such events would be required for the upper limit of 16 events for each explanatory variable. Given a 50% risk of recurrence, a minimum of 160×2=320 subjects will be needed. Allowing for 20% attrition/loss during follow-up given the longitudinal nature of the study, we needed approximately 400 patients, which we set more conservatively at 450.[59–62] However, we do not expect a multivariable model to exceed 10 predictors. There is the possibility of differences between populations recruited during the two phases of the study. If this is not the case, an indicator variable which annotates the 2 phases will be added to the set of predictors and 450 patients would still allow at least 80% power under the conditions described. For example, if 11 predictors are considered, 352 patients would be required or a total of 440 allowing for 20% attrition. If there are substantial differences between subjects recruited in the 2 phases, analyses will consider up to 10 events per explanatory variable requiring 100 events per phase among 200 participants followed up or 250 recruited (allowing for 20% attrition). The latter scenario will require recruitment of a total of 500 during the two phases, a target that we are confident we can achieve.

The definition of an asthma attack in both phases of the study is an episode of acute wheezing requiring an unscheduled visit to a health professional (including visits to ERs) and which is controlled by $\beta_2$-agonist bronchodilator treatment. The original definition for an asthma attack in phase I that included ER attendance had to be modified in the light of changes in health-seeking behaviour caused by the COVID-19 pandemic and fear of contagion in hospitals and other acute health facilities.[19] Potential explanatory variables include age, gender, ethnicity, socioeconomic status, medication use, asthma attacks and use of systemic corticosteroids in year prior to recruitment, asthma control test score and study centre. Given changes to the phase I protocol, an additional variable (for recruitment from ERs vs hospital lists) will be considered as an effect modifier. Subgroup analyses of phase II will include SARS-CoV-2 seropositivity as an explanatory variable. Secondary outcomes will be asthma control score and economic costs to the patient's family.

## Statistical analyses

The statistical models to be used by study objective are summarised in table 1. The primary outcome will be any asthma attack recurrence during 12 months of follow-up in phase I and at least 6 months in phase II. Logistic and linear regression and survival analyses will be used. For the latter, if no reattendance related to an asthma attack occurred during the follow-up period, the patient is considered (right) censored. The general assumption for time to event (survival analysis) is that the censoring is non-informative, that is, it is independent of the events (asthma attacks). Under this assumption, the results should not be qualitatively different from those inferred from a logistic regression. ORs and/or HRs and their corresponding p values and 95% CIs will measure the associations between asthma attack recurrence and explanatory variables, and their statistical significance and uncertainty. The Cox proportional hazard semiparametric technique allows no assumption to be made on the baseline hazard; nevertheless, proportionality of the hazards should hold. The latter assumption can be tested on the basis of Shoenfeld residuals after fitting models to the data. If unsatisfactory, alternative parametric models will be considered after understanding the distributional assumptions on the data such that the baseline hazard function is modelled appropriately. After exploring the data as described above, a forward–backward elimination survival analysis would be applied to determine the most parsimonious model for an adjusted multivariable model; that is, the model with the least number of predictors yet explaining most of the variability in the data. Penalised-likelihood criteria such as Akaike information criterion or Bayesian information criterion will be used to choose between non-nested models—nevertheless based on complete observations only.

Hosmer-Lemeshow measure of goodness of fit would be employed for logistic regression models with a p value of less than 0.05 indicating a poor fit to the data. Model validation will be done by splitting the data set into training–testing subsets (logistic regression setting). Cross validation techniques will be used to understand the prediction power of the corresponding logistic regression model using area under the receiver operator curve, prior to propose the model entering clinical practice with the view of elaborating a prognostic model. In a survival setting, it is also possible that the precise time of the attacks may be missed but recorded as between two time points. This would fall into interval censored data as an alternative to survival setting in which the precise time of the event is required.

Multiple recurrences of asthma attacks within the follow-up period will generate a longitudinal binary outcome, with multiple records for each participant over follow-up. The hierarchical structure of the data requires special inferential techniques, which account for the two sources of variability: that of between participants and within participants. Three different modelling approaches will be considered: generalised estimation equation, mixed models and random effects survival models. The first approach provides population average estimates, the second provides subject-specific estimates, while the third is also known as shared frailty survival model. A frailty is a latent random effect, which enters multiplicatively on the hazard function and accounts for the multiple failures setting and/or for the clustered aspect of the data.

Complete data analyses will be done under missing completely at random assumptions. However, patterns in missing data will be explored and methods to evaluate potential impact of missing data on estimates will be used as appropriate such as multiple imputations using chained equations accompanied by a sensitivity analysis to the missing data assumptions.[63–65] Analyses will use STATA (StataCorp. 2017. Stata Statistical Software: Release V.15. College Station, Texas: StataCorp) or R (R Core Team (2013)).

## DISCUSSION

The prevalence of childhood asthma has increased in many LMICs over the past 50 or so years where asthma attacks are a growing burden to health systems.[2] There are limited data on determinants of asthma attacks in children and adolescents from LMIC settings,[18] and a need for prospective studies to identify risk factors for asthma attacks so that the limited resources available in low-resource settings can be focused on those children most likely to suffer recurrent attacks to reduce morbidity and economic costs to health systems and patients' families. The Asthma ATTACK Study aims to identify risk factors associated with asthma attacks among children and adolescents attending ERs in public hospitals in three Ecuadorian cities.

This study provides a range of climatic settings in Ecuador from high altitude (Cuenca and Quito) to sea level (Portoviejo). Study activities were interrupted in March 2020 by the COVID-19 pandemic resulting in a suspension of recruitment in hospital ERs owing to dramatic changes in access to and use of health facilities. As a consequence, the study protocol was modified to allow reinitiation of recruitment and also the evaluation of the effect of exposures to SARS-CoV-2 on risk of asthma attacks and symptoms in children. The study will help fill an unmet knowledge gap on the effects of SARS-CoV-2 on paediatric asthma, particularly from low-resource LMIC settings.[19]

### Study limitations

A potential limitation is recruitment of sufficient subjects with an acute asthma attack attending ERs at the three public hospitals and ensuring monthly follow-up in phase I of the study. The COVID-19 pandemic had a major impact on health-seeking behaviours while mitigation strategies to control transmission of the novel coronavirus, SARS-CoV-2, resulted in marked declines in the circulation of respiratory viruses that are considered to cause a high proportion of asthma attacks.[66] A shift to telemonitoring during the COVID-19 pandemic helped ensure almost complete follow-up of those recruited prior to March 2020, although procedures requiring the presence of the patient or that generated potentially hazardous aerosols (eg, spirometry) had to be suspended. Declines in circulating respiratory viruses[67] because of mitigation strategies against COVID-19 could reduce the number of recurrence events and hence the power of the study. Phase II is recruiting known asthmatic children and adolescents with recent asthma symptoms and following them up for asthma attacks for at least 6 months. The change in criteria should allow sufficient subjects to be recruited given the impact of COVID-19 pandemic on health attendance behaviours, although continuing mitigation measures may reduce recurrence. Follow-up in phase II is being done through telemonitoring and a few scheduled face-to-face visits either to the study clinic or the participant's home, depending on individual preferences. Other potential limitations include missing of events, although monthly contacts should minimise these. Significant losses to follow-up that could lead to selection bias are being minimised by regular contacts with caregivers including home visits, and confounding should be reduced by the collection of data on a wide variety of known potential risk factors and confounders including treatments received and adherence to those treatments.

### Ethics and dissemination

Ethical approval for phase I was obtained from the Ethics Committee of the Hospital General Docente de Calderon (CEISH-HGDC 2019-001) and for phase II from the Ecuadorian Ministry of Public Health COVID-19 Ethics Committee (MSP-CGDES-2021-0041-O N° 096-2021). The study results will be disseminated through presentations at conferences and to key stakeholder groups including policy-makers, postgraduate student theses, peer-review publications and on a study website. Participants gave informed consent to participate in the study before taking part.

**Author affiliations**
[1]Escuela de Medicina, Universidad Internacional del Ecuador, Quito, Ecuador
[2]Department of Biosciences, Universidad de Cuenca, Cuenca, Ecuador
[3]Facultad de Medicina, Universidad de Azuay, Cuenca, Ecuador
[4]School of Medicine, Universidad San Francisco de Quito, Quito, Ecuador
[5]Emergency Department, Hospital General Docente Calderón, Quito, Ecuador
[6]Emergency Department, Hospital Verdi Cevallos Balda, Portoviejo, Ecuador
[7]Pediatric Pneumology, Hospital de Especialidades Portoviejo, Portoviejo, Ecuador
[8]Instituto de Ciências da Saúde, Universidade Federal da Bahia, Salvador, Bahia, Brazil
[9]Institute of Social and Preventive Medicine, University of Bern, Bern, Switzerland
[10]Norwich Medical School, University of East Anglia, University of East Anglia, Norwich, UK
[11]Population Health Research Institute, St George's, London, UK
[12]Institute of Infection and Immunity, St. George's University of London, London, UK
[13]Núcleo de Excelência em Asma, Universidade Federal da Bahia, Salvador, Brazil
[14]School of Medicine, Universidad Internacional del Ecuador, Quito, Ecuador
[15]GRAAL, Grups de Recerca d'America i Africa Llatines, Cerdanyola del Valles, Barcelona, Spain

**Contributors** Study design—PC, AC, NCR, MB, MRP, CA-G. Study conduct—PC, NCR, DM, SM-B, AO, MEC, CR, AM, KA, AR, CF, JA, KS. Manuscript drafting—PC, DM, SM-B. Statistical analysis plan—ICS. Manuscript editing and review—all authors.

**Funding** This work is based on the Asthma Attacks Causes and Prevention Study in Urban Latin America which is funded by the National Institute of Health Research under the Global Health Research Programme (grant 17/63/62). Additional financial support was provided by the Corporación Ecuatoriana para el Desarrollo de la Investigación y la Academia CEDIA (N° CEPRA-XV-2021-03).

**Competing interests** None declared.

**Patient and public involvement** Patients and/or the public were not involved in the design, or conduct, or reporting, or dissemination plans of this research.

**Patient consent for publication** Not applicable.

**Provenance and peer review** Not commissioned; externally peer reviewed.

**Open access** This is an open access article distributed in accordance with the Creative Commons Attribution 4.0 Unported (CC BY 4.0) license, which permits others to copy, redistribute, remix, transform and build upon this work for any purpose, provided the original work is properly cited, a link to the licence is given, and indication of whether changes were made. See: https://creativecommons.org/licenses/by/4.0/.

**ORCID iDs**
Max Bachmann http://orcid.org/0000-0003-1770-3506
Michael Richard Perkin http://orcid.org/0000-0001-9272-2585
Irina Chis Ster http://orcid.org/0000-0003-2637-1259
Natalia Cristina Romero http://orcid.org/0000-0001-6881-6581
Philip Cooper http://orcid.org/0000-0002-6770-6871

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
