## [Reviewer comments · BMJ Open]

ARTICLE DETAILS

TITLE (PROVISIONAL)	A prospective study of factors associated with asthma attack recurrence (ATTACK) in children from three Ecuadorian cities during COVID-19: a study protocol
AUTHORS	Morillo, Diana; Mena-Bucheli, Santiago; Ochoa, Angélica; Chico, Martha E; Rodas, Claudia; Maldonado, Augusto; Arteaga, Karen; Alchundia, Jessica; Solorzano, Karla; Rodriguez, Alejandro; Figueiredo, Camila; Ardura-Garcia, Cristina; Bachmann, Max; Perkin, Michael; Chis Ster, Irina; Cruz, Alvaro; Romero, Natalia; Cooper, Philip

VERSION 1 – REVIEW

REVIEWER	Al-Sallakh, Mohammad Swansea University Medical School
REVIEW RETURNED	29-Oct-2021

GENERAL COMMENTS	This is a protocol for a prospective study about identifying risk factors of asthma attacks recurrence among children in Ecuador. The protocol is well written and the background and methods are adequately detailed. I recommend publication after correcting typos (e.g., page 9, first line in "Recruitment -Phase I").
---

REVIEWER	Asamoah-Boaheng, Michael Memorial University of Newfoundland
REVIEW RETURNED	14-Mar-2022

GENERAL COMMENTS	1) The authors need to spell out the number of specific objectives they intend to answer in phases 1 and 2 of their research and how they intend to achieve each specific objective. 2) The choice of the sample size in this study proposal needs to be justified. The authors need to justify the choice of their sample size by providing further details about how the sample size quoted in the protocol was estimated. This can be done through manual calculation or using software 3) Also, the authors need to indicate any anticipated limitations of this study. 4) Also, since the authors will be measuring the effects of risk factors for asthma recurrence over time, they should spell out clearly, the specific parametric survival models that will be used to answer each
---

	of the research questions in the proposal. In other words, the authors should explain how each of the study's objectives will be addressed in the proposal and what specific statistical methods will be used to compute the effects estimates of the respective research objectives.
--	--

VERSION 1 – AUTHOR RESPONSE

Reviewer: 1

Dr. Mohammad Al-Sallakh, Swansea University Medical School

Comments to the Author:

This is a protocol for a prospective study about identifying risk factors of asthma attacks recurrence among children in Ecuador.

The protocol is well written and the background and methods are adequately detailed. I recommend publication after correcting typos (e.g., page 9, first line in "Recruitment -Phase I").

Response 4: manuscript has been reviewed for typos.

Reviewer: 2

Dr. Michael Asamoah-Boaheng, Memorial University of Newfoundland

Comments to the Author:

1) The authors need to spell out the number of specific objectives they intend to answer in phases 1 and 2 of their research and how they intend to achieve each specific objective.

Response 5: Information on study objectives and how we plan to achieve these objectives is provided in a new Table 1

2) The choice of the sample size in this study proposal needs to be justified. The authors need to justify the choice of their sample size by providing further details about how the sample size quoted in the protocol was estimated. This can be done through manual calculation or using software

Response 6: Additional information on sample size estimation is now provided on page 17.

3) Also, the authors need to indicate any anticipated limitations of this study.

Response 7: A limitations section is now provided in the Discussion on page 20.

4) Also, since the authors will be measuring the effects of risk factors for asthma recurrence over time, they should spell out clearly, the specific parametric survival models that will be used to answer each of the research questions in the proposal. In other words, the authors should explain how each of the study's objectives will be addressed in the proposal and what specific statistical methods will be used to compute the effects estimates of the respective research objectives.

Response 8: We have produced a table (new Table 1) of potential outcomes in the data, how these outcomes will be addressed, and information on statistical analyses (outcomes, models, and inference) for each objective that we envisage that could apply to these data. The statistical analysis section has been edited in line with the information provided in Table 1.

VERSION 2 – REVIEW

REVIEWER	Asamoah-Boaheng, Michael Memorial University of Newfoundland
REVIEW RETURNED	12-May-2022
GENERAL COMMENTS	The authors addressed all my comments